# Development of Mathematical Model to Predict Soymilk Fouling Deposit Mass on Heat Transfer Surfaces Using Dimensional Analysis

Eakasit Sritham [1], Navaphattra Nunak [1,*], Ekarin Ongwongsakul [1], Jedsada Chaishome [1], Gerhard Schleining [2] and Taweepol Suesut [1]

1   School of Engineering, King Mongkut's Institute of Technology Ladkrabang, Bangkok 10520, Thailand; eakasit.sr@kmitl.ac.th (E.S.); 65016105@kmitl.ac.th (E.O.); jedsada.ch@kmitl.ac.th (J.C.); taweepol.su@kmitl.ac.th (T.S.)
2   Department of Food Science and Technology, University of Natural Resources and Life Sciences, Vienna (BOKU), 1190 Vienna, Austria; gerhard.schleining@boku.ac.at
*   Correspondence: navaphattra.nu@kmitl.ac.th

**Abstract:** The formation of fouling deposits on heat exchanger surfaces is one of the major concerns in thermal processes. The fouling behavior of food materials is complex, and its mechanism remains, in general, unclear. This study was aimed at developing a predictive model for soymilk fouling deposit formed on heated surfaces using dimensional analysis. Relevant variables affecting fouling deposit mass could be grouped into six dimensionless terms using Buckingham's pi-theorem. Experimental data were obtained from a lab-scale plate heat exchanger. A simple model developed using the experimental data under the process conditions with the product inlet temperature, the product outlet temperature, and plate surface temperature in the ranges of 50–55 °C, 65–70 °C, and 70–85 °C, respectively, exhibited a good performance in the prediction of soymilk fouled mass. The correlation coefficient between the predicted and experimental values of fouled mass was 0.97 with an average relative error of 9.03%. Within the ranges of product inlet temperature and plate surfaces temperature studied, this model offers an opportunity to estimate soymilk fouling mass with acceptable accuracy.

**Keywords:** fouling; soymilk deposit; dimensional analysis; fouling mass model



## 1. Introduction

Pasteurization with a plate heat exchanger (PHE) is commonly used in the dairy milk and plant-based milk industry to ensure product safety, quality, and extending product shelf life [1]. In addition, heating could also eliminate off-flavors of plant-based milk products such as soymilk [2,3], oat milk, almond milk, and other milk produced from several varieties of nuts. During thermal processing, protein in milk products tend to denature, aggregate, or interact with other components, leading to deposit formation on contact surfaces [4]. The fouling formation within the range of pasteurization temperature from 75 to 100 °C contains mainly protein denaturation, which is a combined effect of unfolding and aggregation phenomena [5]. The protein denaturation could generate on the heat exchanger surface and build up deposits. Deposit formation, also known as fouling, is a severe problem in industrial processes including food processing plants as it can decrease the efficiency of heat exchangers [6] and increase the pressure drop in the system. Once the fouling occurs, the production line needs to be stopped for cleaning to recover the initial operating condition.

The fouling behavior of protein in milk during thermal processing is highly complex [7]; there are many variables affecting the fouling of protein on heating surfaces. These variables can be divided into three main groups, namely product variables (e.g., concentration [8–10], protein type [11], protein reactivity, denaturation, aggregation, and

deposition temperatures [12,13], physiological, thermal, and rheological properties [9]), process variables (e.g., flow velocity, inlet and outlet temperatures of product heating medium or surface temperature, and processing time [8,9,14]), and PHE configuration or equipment design variables (e.g., plate surface area and space between two plates [9]). Temperature is a key factor that strongly affects the thermal denaturation of protein leading to deposition of the protein on contact surface or fouling [11,13,15]. Belmar-Beiny et al. [14] have reported a nonlinear relationship between protein deposit mass formed on contact surface and temperature. Protein fouling behavior is complex, and the development of theoretical basis prediction model of this behavior requires thorough understanding of its mechanisms. For decades, a number of researchers have studied and tried to develop fouling models. Pan et al. [11] applied two-dimensional computational fluid dynamics (2D-CFD) incorporated with bulk and surface reaction kinetics for β-lactoglobulin (β-lg) and momentum, mass, and heat transfers to develop a predictive model for dairy milk fouling. The investigation using the obtained model revealed a two-stage growth behavior of fouling deposit layer on the contact surface of heat exchanger over the heating period. Indumathy et al. [16] proposed a one-dimensional fouling dynamic model of a high-temperature–short-time pasteurization process (HTST), called a plant model, to simulate temperature profiles at the exit of pasteurization process. This model was developed using the conservation of mass and energy principles together with the protein fouling model and validated with experimental data collected from the plant. Model improvement was then made with the log-mean-temperature-difference (LMTD) approach. This dynamic plant model could be used to simulate the outlet temperature of milk at the heating section, which is closely related to the formation of protein fouling layer. The outlet temperature of milk was used as a key parameter to control the inlet temperature of heating medium. However, the robustness of CFD and theoretical models are generally limited by a number of model assumptions and appropriate boundary conditions are difficult to assign. Moreover, the coupling of relevant fundamental transport equations and the physical, chemical, or biological kinetics is complex [17,18], which makes it difficult to interpret.

Fryer et al. [8] developed a statistical model using multiple linear regression (MLR) to correlate the data—e.g., inlet protein concentration, product flow rate, the temperature difference across the test section, and the amount of protein reacted. It was found that kinetics of denaturation and aggregation of whey protein solution within an ultra-high-temperature (UHT) process could be related to the degree of fouling and could be used to predict the heat transfer coefficients of heat exchangers. The ability of statistical models to predict a target variable depends on the size of data; a large number of experiments are generally required for the development of reliable models. Collecting such a great deal of data could consume a huge amount of resources—e.g., time, cleaning agent, electricity, energy, and raw material [18]. While the development of CFD, theoretical, and statistical models involves various limitations, a dimensional analysis (DA) technique can be used to establish a model without in-depth knowledge of the process studied. By the application of DA, several dimensional variables can be grouped into fewer dimensionless numbers. Not only can the DA help reducing the number of model parameters and complexity, but the number of experiments can be minimized [9,18]. Nowadays, DA is commonly used to develop the model for prediction of fouling deposit mass on heat transfer surfaces, especially in a case where reaction kinetics plays major role in the fouling mechanism. Petit et al. [12] applied the DA to identify the process relationships among β-lg denaturation level, aggregate size, fouling mass, inlet temperature and reactivities of β-lg concentrate, and hot water inlet temperature. It was found that the Arrhenius exponential factor of β-lg of unfolding reaction could be used to describe the denaturation, aggregation, fouling mechanism and used to be the guideline for controlling β-lg aggregation and fouling during heat treatments. Gu et al. [9] applied the DA technique by taking into account the composition and physicochemical properties, and the temperature of whey protein solutions to develop a predictive model for whey protein fouling mass in a PHE during thermal treatment. It was found that the obtained simple model helped to gain a better

understanding of the fouling developed in the PHE within the parameter ranges studied, which would otherwise have a large number of variables and require deep knowledge of reaction kinetics. In another study by Georgiadis et al. [19], the authors developed a model using the 2D-CFD to predict the fouling behavior of dairy milk within the PHE. Though the model could describe thermal and hydraulic performances, prediction of fouling mass was limited due to the declaration of assumptions and the difficulty to identify the parameters used for the models. Alhuthali et al. [10] recently proposed a technique to expand the applicable range of theorical-based models for capturing the fouling growth of whey protein using the DA. The adjusted fouling model showed the capability of the total mass and the deposit mass per channel for a wider range of product and process conditions even with a limited amount of experimental data used for the development.

It can be clearly seen in the literature that the temperatures of product and contact surface are highly influential variables that affect milk protein unfolding and deposition on hot surfaces. Nowadays, plant-based milk has gained popularity in consumer markets worldwide. This has brought about the need for research in a multitude of aspects to support the growth of the plant-based milk industry. The cleaning of processing lines could take up a large part of production costs as it involves time, cleaning agents, and other process utilities. Consequently, process efficiency may be improved by optimizing the cleaning protocol. An effective cleaning protocol requires a thorough understanding of the fouling behavior of the product. In addition, knowledge about fouling behavior would be also beneficial to PHE manufacturers. A better understanding of the behavior of fouling process may be gained though the study of fouling models.

Among a variety of plant-based milk products, soymilk has long stood out as one of the most favorable beverages. However, information about the thermal denaturation of proteins in soymilk is still very limited, making it difficult to develop an applicable prediction model for fouling behavior based on theoretical or statistical approaches. This study aimed to apply the DA technique to develop dimensionless groups among relevant variables of fouling and obtain a simple model for the prediction of soymilk fouling deposit mass formed on contact surfaces of PHE in the pasteurization temperature range.

## 2. Materials and Methods

### 2.1. Sample Preparation

Soymilk samples were freshly prepared before performing the experiments. Each sample was made with 1000 g of soybeans. The beans were washed, soaked in distilled water for 12 h at room temperature (25 ± 2 °C), drained, and rinsed. The beans were then ground at room temperature using a soymilk grinder, and distilled water was continually added during grinding. Soybean residue was automatically separated from raw soymilk. The total solid of raw soymilk samples was approximately 15% ($w/w$).

### 2.2. Experimental Set-Up

A laboratory-scale high-temperature, short-time (HTST) pasteurizer (FT75 Laboratory Pasteuriser, Armfield, UK) with some modifications was used in this study. Heat treatment was made in the heat exchanger section of the pasteurizer. The modification was made on the sample and water feeding system. A schematic diagram of the experimental set-up is given in Figure 1, a holding tank, two 22 L water baths (Memmert, Germany, an accuracy of ± 0.3 °C), two pumps, fluid supply lines, and a temperature-control sub-system. The heating section consisted of 7 corrugated heat transfer plates made from stainless steel of approximately 0.1 mm thickness. The plates were arranged for the countercurrent flow of two fluids in one channel-per-pass. The projected area of each corrugated heat transfer plate ($S_0$) was 86.3 cm$^2$ ($L_w \times L_h$; 75 mm × 115 mm) and total heat exchanging projected surface area ($S_{exch}$) was 431.5 cm$^2$. The average space between 2 plates ($b$) is 3 mm. Hot water was used as a heating medium. A holding tank contained 8 L of soymilk sample. Thermocouple sensors (K-type) were installed in storage tanks and a holding tank to record and monitor the inlet and outlet temperatures of water and soymilk every 30 s during the

experiments. Heat loss from the system was neglected and so the flowing fluid system was assumed to be isothermal.

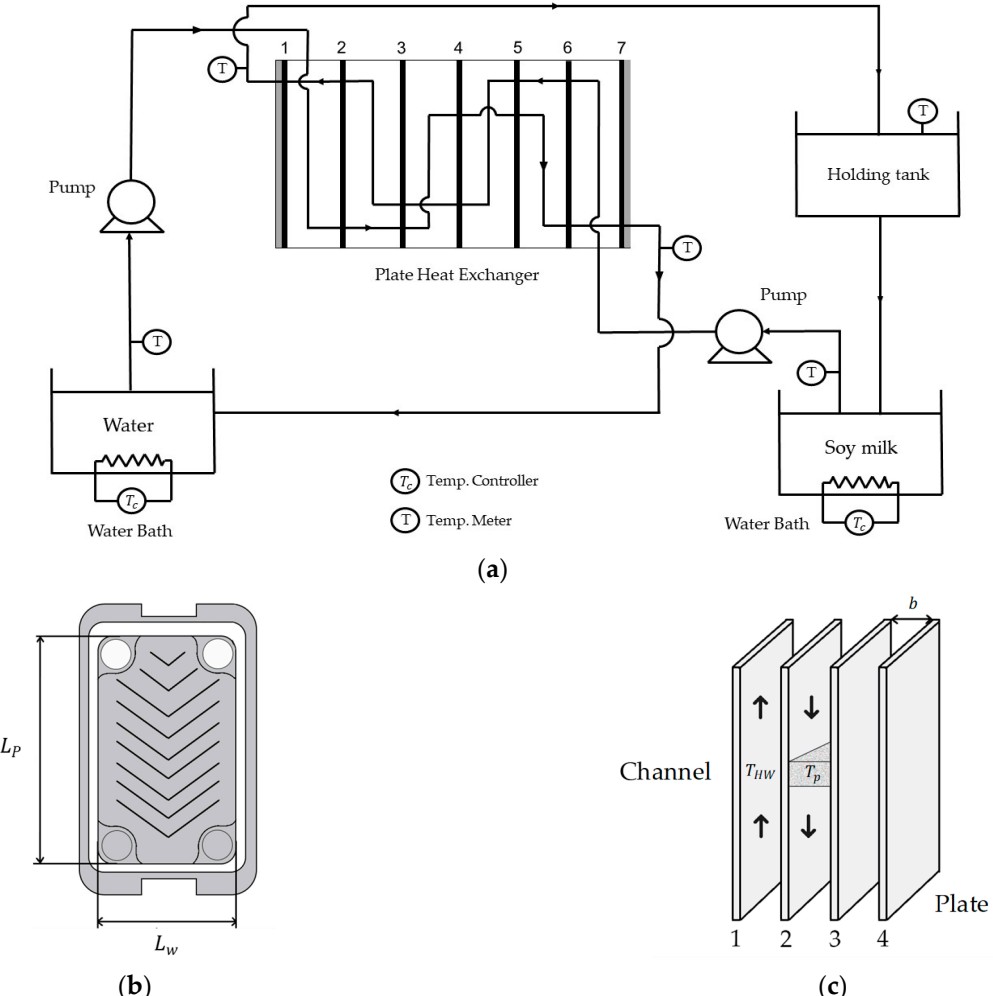

**Figure 1.** (**a**) Schematic diagram of the fouling test apparatus; (**b**) dimension of heat exchanger plate; (**c**) product (downward arrow) and heating medium (upward arrow) flow directions.

### 2.3. Fouling Experiments

The effect of fouling surface temperature ($T_s$) was studied in a range of 65–85 °C. The $T_s$ was controlled by recirculating hot water through the PHE at a constant velocity of 0.408 m s$^{-1}$. The $T_s$ was found to be in a range of ±1.5 °C from hot water temperature. A water bath with temperature control unit was used to preheat soymilk from room temperature to 45 °C. Soymilk was circulated through the pasteurizer using a peristaltic pump to achieve the desired inlet temperature ($T_{pi}$) of 50 °C and 55 °C after which the production run time was started. The density ($\rho$) and dynamic viscosity ($\mu$) of soymilk at an average temperature of 65 °C were 1045 kg m$^{-3}$ [20] and 0.038 kg m$^{-1}$s$^{-1}$ [21], respectively. The velocity of soymilk in the channel of heat exchanger ($v$) was kept constant at 0.048 m s$^{-1}$, which corresponded to the Reynolds numbers ($Re$) of 7.92, indicating the laminar flow regime. In each fouling experiment, soymilk was recirculated through the experimental setup for approximately 70 min. The fouling deposit mass was assumed to form on the heating surface after the first 10 min. Therefore, the heating time ($t$) for each test run was 60 min.

After finishing each test run, the plates of heat exchanger were dismantled from the frame and put on a table in an inclined position of 30 degrees at ambient temperature for 5 min. Then, individual fouled plates were weighed with a digital balance (Vibra-AB323, Shinko Denshi, China, a resolution of 0.001 g, and an accuracy of ±0.003 g). The fouling

deposit mass ($m_f$) was then calculated by subtracting the mass of fouled plat with that of dry-cleaned plate. The deposit mass formed on the 7th plate was the fouling of unheated surface by soymilk. Therefore, total fouling deposit mass was collected from the 2nd to 6th plates.

*2.4. Dimensional Analysis of Fouling Deposit Mass*

The general scheme of dimensional analysis (DA) includes listing the related physical variables, constructing, and rearranging dimensionless numbers, and analyzing the process relationships. In this study, there were nine variables, with four fundamental dimensions, namely mass (M), length (L), time (T), and temperature (K), which affected the fouling deposit mass formed on the PHE—the target variable. It should be noted that deep understanding of the mechanisms and reaction kinetics of the fouling was not applied here. That means DA allows the development of a simple prediction model without in-depth knowledge about fouling phenomenon. The chosen variables (given in Table 1) were only practical product, process, and design parameters.

**Table 1.** Selected variables for the DA of soymilk fouling deposit mass in the PHE.

| Type of Variable | Symbol | Variable Name | Unit | Dimension |
|---|---|---|---|---|
| Target variable | $m_f$ | Fouling mass | kg | $M^1$ |
| Physical variables<br>Product | $\rho$ | Density of product [1] | kg m$^{-3}$ | $M^1L^{-3}$ |
| | $\mu$ | Dynamic viscosity of product [1] | kg m$^{-1}$s$^{-1}$ | $ML^{-1}T^{-1}$ |
| Process | $T_s - T_{po}$ | The difference of surface temperature and the product outlet temperature | K | $K^1$ |
| | $v$ | Velocity of product [1] | ms$^{-1}$ | $LT^{-1}$ |
| | $\Delta T_p = T_{po} - T_{pi}$ | The difference of inlet and outlet temperatures of product | K | $K^1$ |
| | $t$ | Heating time [1] | s | $T^1$ |
| Design | $D_h$ | Hydraulic diameter [1] | m | L |
| | $S_{exch}$ | Total heat exchanging projected surface area [1] | m$^2$ | $L^2$ |
| | $S_0$ | Projected surface area per plate [1] | m$^2$ | $L^2$ |

[1] Fixed variables as explained in Sections 2.2 and 2.3.

The dimensional matrix of all variables and their dimensions is presented in Table 2. The Vaschy–Buckingham theorem ($\pi$-theorem) was applied to a set of physical variables to determine the maximum dimensionless numbers. All physical variables were divided into a set of non-repeated variables and a set of repeated variables. The repeated variables must be selected so that their dimensions are independent and cover all the fundamental dimensions of all relevant physical variables. Accordingly, the number of repeated variables shall be equal to the number of fundamental dimensions [18]. Therefore, in this study, four variables including the hydraulic diameter of a flow channel ($D_h$), the velocity of soymilk ($v$), the density of soymilk ($\rho$), and the difference between surface temperature and the product outlet temperature ($T_s - T_{po}$), were chosen as the repeated variables.

**Table 2.** Dimensional matrix of target and physical variables.

| | $m_f$ | $t$ | $\Delta T_p$ | $\mu$ | $S_0$ | $S_{exch}$ | $D_h$ | $v$ | $\rho$ | $T_s - T_{po}$ |
|---|---|---|---|---|---|---|---|---|---|---|
| K | 0 | 0 | 1 | 0 | 0 | 0 | 0 | 0 | 0 | 1 |
| M | 1 | 0 | 0 | 1 | 0 | 0 | 0 | 0 | 1 | 0 |
| L | 0 | 0 | 0 | −1 | 2 | 2 | 1 | 1 | −3 | 0 |
| T | 0 | 1 | 0 | −1 | 0 | 0 | 0 | −1 | 0 | 0 |

Following Buckingham's pi-theorem, the ten variables—target and physical variables—were grouped into six dimensionless numbers by subtracting the number of variables (10 variables) with the fundamental dimensions (4 dimensions) as given in Equation (1).

$$\pi_1 = f(\pi_2, \pi_3, \pi_4, \pi_5, \pi_6) \tag{1}$$

The dependent $\pi$ term ($\pi_1$), fouling deposit mass, can be expressed as a function of dimensionless groups as given in Equation (2).

$$\pi_1 = \frac{m_f}{D_h{}^3 \rho} = f\left(\pi_2 = \frac{tv}{D_h}, \pi_3 = \frac{(T_{po} - T_{pi})}{(T_s - T_{po})}, \pi_4 = \frac{\mu}{\rho v D_h}, \pi_5 = \frac{S_0}{D_h{}^2}, \pi_6 = \frac{S_{exch}}{D_h{}^2}\right) \tag{2}$$

The term $\pi_4$ was then rearranged by multiplying the exponent with $-1$, yielding the $\pi_4{}'$ known as Reynolds number as presented in Equation (3).

$$\pi_4{}' = \frac{\rho v D_h}{\mu} = Re \tag{3}$$

Since $D_h$ is twice the size of the average space between two plates ($b$), Equation (3) will be rewritten as given in Equation (4).

$$Re = \frac{\rho v 2b}{\mu} \tag{4}$$

The dimensionless numbers for predicting the fouling deposit mass of soymilk are described as follows:

(1)   $\pi_1$ relates to the total fouling deposit mass formed on PHE, density of product, and the hydraulic diameter of the plate;

(2)   $\pi_2$ relates to the duration that the product contacts with the heated plate;

(3)   $\pi_3$ is the ratio of the difference of product outlet and inlet temperatures to the difference of surface and product outlet temperatures;

(4)   $\pi_4{}'$ is Reynolds number;

(5)   $\pi_5$ is the ratio of the projected surface area per plate to the hydraulic diameter of the plate;

(6)   $\pi_6$ is the ratio of the total heat exchanging projected surface area to the hydraulic diameter of the flow channel.

### 2.5. Statistical Analysis

The fouling experiments (54 runs) were conducted using the completely randomized design (CRD) by considering the dimensionless numbers ($\pi$ terms) as experimental variables. Experimental data were equally divided into two parts. The first part one was used for model development, and another part was used for model validation. Nonlinear regression was carried out with a statistical package (IBM SPSS Statistics V.28). All statistical analyses were made at 0.05 level of significance.

## 3. Results and Discussion

### 3.1. Predictive Model for Soymilk Fouling Deposit Mass on Heated Surface

The development of a predictive model in this study provided relationships among relevant variables in terms of six dimensionless numbers that could be beneficial for further study. Since the temperatures of product and heated surface are the most influential factors, the process in this experiment was performed so that the terms $\pi_2 = \frac{tv}{D_h}$, $\pi_4 = Re$, $\pi_5 = \frac{S_0}{D_h{}^2}$, and $\pi_6 = \frac{S_{exch}}{D_h{}^2}$ were kept constant at 28,800, 7.92, 238.88, and 1198.61, respectively. The configuration of the system and operating conditions are shown in Figure 2. As the terms $\pi_2, \pi_4, \pi_5$, and $\pi_6$ were fixed, the model could be further simplified by bundling those fixed terms into a single constant term. That said, the configuration of the system was reduced from six to one internal measure. The effect of product and surface temperatures

on the fouling mass was inferred from $\pi_3$. Therefore, the predictive model in Equation (2) was simplified to the form given in Equation (5).

$$\frac{m_f}{D_h{}^3 \rho} = f\left(\frac{(T_{po} - T_{pi})}{(T_s - T_{po})}\right)$$

(5)

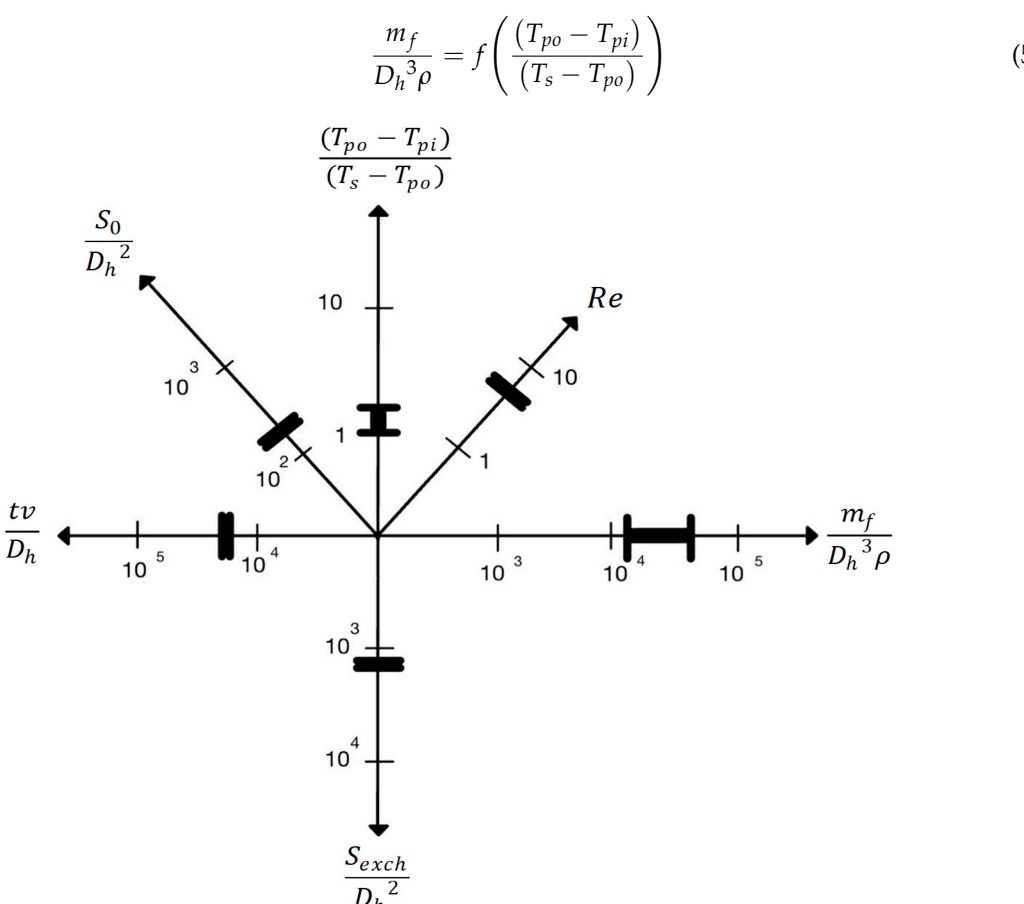

**Figure 2.** The configuration of the system and operating conditions with six internal measures. Black bars with limits at each end are the range of operating points covered by each internal measure individually.

The fouling model to predict the deposit mass of soymilk formed on hot surfaces was developed from the experimental data in the range of $8464.32 \leq \pi_1 \leq 53{,}504.34$ and $1.35 \leq \pi_3 \leq 3.21$. The term of $\pi_3$ was varied by changing the inlet temperature of heating medium and soymilk. Experimental temperature setting and the corresponding product outlet temperature are given in Table 3.

**Table 3.** Experimental temperature setting and the corresponding product outlet temperature.

| Product Inlet Temperature ($T_{pi}$, °C) | Surface Temperature ($T_s$, °C) | Product Outlet Temperature ($T_{po}$, °C) |
|---|---|---|
| 50 | 65 | 60.43 ± 0.34 |
| 50 | 70 | 65.67 ± 0.21 |
| 50 | 75 | 67.07 ± 0.46 |
| 50 | 80 | 70.24 ± 0.05 |
| 55 | 65 | 61.62 ± 0.28 |
| 55 | 70 | 66.85 ± 0.15 |
| 55 | 75 | 69.29 ± 0.24 |
| 55 | 80 | 70.73 ± 0.68 |
| 55 | 85 | 72.25 ± 0.11 |

Data are means ± SD (n = 3).

Total accumulated mass was collected from the second to sixth plates within 1 h after the surface and product inlet temperatures reached the designed levels. Experimental data

on fouling mass as a function of temperature ratio ($\pi_3$) are given in Figure 3. It was found that at the surface temperature ($T_s$) of 65 °C and product inlet temperature ($T_{pi}$) of 50 °C and 55 °C, the outlet temperature of soymilk ($T_{po}$) was approximately 61–62 °C, which is the level of temperature where protein aggregation only begins to develop [22]. These conditions corresponded to the $\pi_3$ term of 2.23 and 2.31, respectively. Close inspection on heated plates revealed that there was only a thin layer of fouled mass presenting on the surface with the total of approximately 1.91 g. On the other hand, at some other experimental conditions where the $\pi_3$ terms were nearly in the same range (2.0–2.3) but the product outlet temperatures were higher (67–70 °C), considerably larger amounts of fouled mass (8.0–9.3 g) were observed. When considering the trend of experimental data in Figure 3, most of the fouled mass data linearly correlated with $\pi_3$ terms, except the experimental conditions where product outlet temperature was only 61–62 °C. Consequently, the data obtained from these experimental conditions (marked red, in Figure 3) were excluded from subsequent model development.

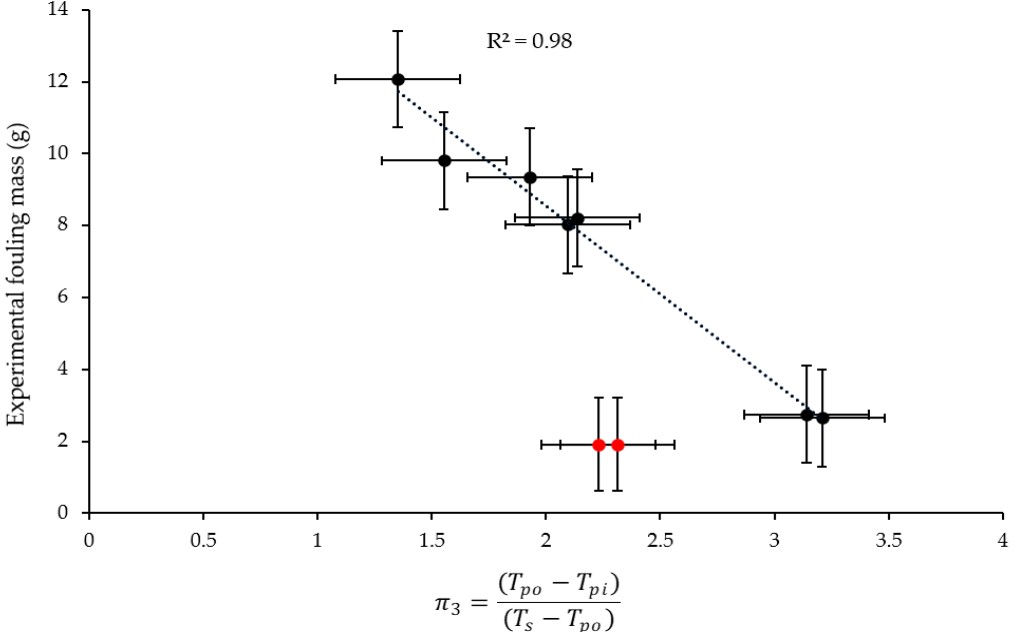

$$\pi_3 = \frac{(T_{po} - T_{pi})}{(T_s - T_{po})}$$

**Figure 3.** The evolution of total soymilk fouled mass formed on heat exchanger plates at various dimensionless process parameters. Data are means and error bars are standard deviation (n = 3). A regression line was obtained with the data marked in black. Operating conditions corresponding to the data marked in red were excluded from the model development.

It was observed that, in general, $m_f$ was dependent on product inlet and surface temperatures, as could be clearly observed from the trend in Figure 3, where $m_f$ tends to decrease linearly with $\pi_3$. The term in the left-hand side of Equation (5) could be written as $m_f = \pi_1\left(D_h{}^3\rho\right)$. Since $D_h$ and density were treated as constants, it could be inferred from Figure 3 that $\pi_1$ highly correlated with $\pi_3$ ($r^2 = 0.98$).

The relationship among process parameters in terms of $\pi_1$ and $\pi_3$ was then analyzed from the data obtained from the experimental conditions with 50 °C $\leq T_{pi} \leq$ 55 °C and 70 °C $\leq T_s \leq$ 85 °C allowing the product outlet temperature to fall to the range of 65 °C to 70 °C. These experimental conditions corresponded to the range of $\pi_3$ values from 1.35 to 3.21, and the obtained $\pi_1$ ranged from 11,719.53 to 53,504.34. Then, the process relationship was statistically analyzed using nonlinear regression yielding the model parameter values shown in Table 4.

**Table 4.** Results of nonlinear regression analysis for the fouling deposit mass.

| $\pi_3$ | Exponent | Standard Error | *p*-Value |
|---|---|---|---|
| $\dfrac{(T_{po}-T_{pi})}{(T_s-T_{po})}$ | 0.053 | 0.208 | <0.001 |

The resulting predictive model of soymilk fouling deposit mass as a function of the product inlet and outlet temperatures, and the heated surface temperature is presented in Equations (6) and (7).

$$\pi_1 = \left[ -827,413.79 \cdot (\pi_3)^{0.053} \right] + 891,754.96 \tag{6}$$

$$m_f = \left[ -186.76 \cdot \left( \frac{(T_{po}-T_{pi})}{(T_s-T_{po})} \right)^{0.053} \right] + 201.29 \tag{7}$$

*3.2. Effect of Temperature*

The effects of heat exchanger surface temperature and the product inlet temperature on soymilk fouling deposits that occurred during the thermal processing were investigated. Under the temperature conditions applied in this study, the effect of fouling surface temperature was evaluated at five different levels of temperature including 65, 70, 75, 80, and 85 °C and product inlet temperatures of 50 and 55 °C. Figure 4 shows images of wet soymilk fouling formed on the heat exchanger plates after 60 min at different temperature conditions. The higher the temperatures of the heated surface and of the inlet soymilk, the higher the amount of soymilk deposit developed on the plates.

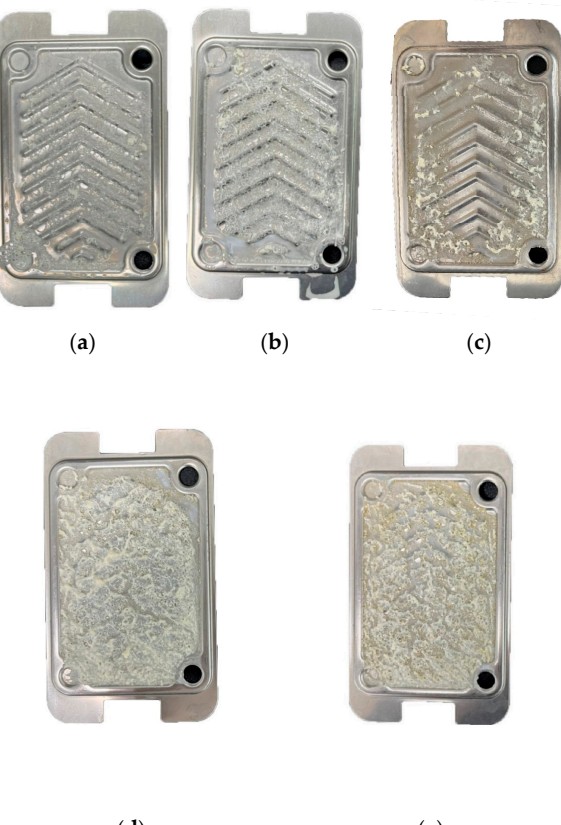

**Figure 4.** Typical images of soymilk fouling deposit mass on the 2nd plate after 60 min at different experimental conditions. (**a**) $T_{pi}$ = 50 °C, $T_s$ = 65 °C. (**b**) $T_{pi}$ = 50 °C, $T_s$ = 70 °C. (**c**) $T_{pi}$ = 50 °C, $T_s$ = 75 °C. (**d**) $T_{pi}$ = 55 °C, $T_s$ = 80 °C. (**e**) $T_{pi}$ = 55 °C, $T_s$ = 85 °C.

The fouling mass tended to increase with the increasing surface temperature. The dramatic increase in fouling mass was observed when the surface temperature was higher than 75 °C at a soymilk inlet temperature of 55 °C. At these temperature conditions, the proteins in soymilk would have already denatured and formed a fouling deposit mass on the surface. Iwabuchi and Yamauchi [23,24] have reported that during the thermal processing of soymilk, β-conglycinin and glycinin, which are the two major components of soy proteins and sensitive to heat treatment, may affect the thermal efficiency of the heat exchanger and play an important role in the formation of fouling deposit. At a temperature above 65 °C, β-conglycinin begins to denature and interact with other proteins to form deposits on a surface. At higher temperatures (above approximately 80 °C), glycinin plays a major role in soymilk deposition [25,26]. Qingjun et al. [27] studied the fouling behavior of soymilk in plate heat exchanger at different inlet temperatures (30, 50, and 70 °C) and found that the fouling factor of soymilk increased with the increasing product inlet temperature. The soymilk fouling mass increased at every increment of soymilk inlet temperature. These fouling characteristics were in good agreement with those reported by Wang et al. [15], Zhang and Xu [28], and Chen and Bala [29] who studied the fouling deposit behavior of soymilk, camel milk, and dairy milk, respectively.

### 3.3. Prediction Performance of the Developed Model

The predicted total mass of wet soymilk deposits formed on the heat exchanger plates was calculated from the term $\pi_1$ which is the response of the model (Equations (6) and (7)), within a range of $11{,}719.53 \leq \pi_1 \leq 0.34$ and $1.35 \leq \pi_3 \leq 3.21$. The plot between the predicted and experimental values is given in Figure 5. Even though this model was developed using limited number of experimental data, it can be seen that the predicted values were in good agreement with the experimental values of soymilk fouling deposit mass with a correlation coefficient (*r*) of 0.97. Figure 6 shows the relative error of the predictions, which were within 9.50% with a mean relative error of 9.03%; this level of relative error was considerably low compared to the literature values [10,12]. The findings provided an opportunity to estimate fouling mass using the model developed with DA approach. The robustness of this model may be improved and verified with experimental data from further studies carried out under slightly different experimental conditions, e.g., design of heating plate, physical properties of soymilk.

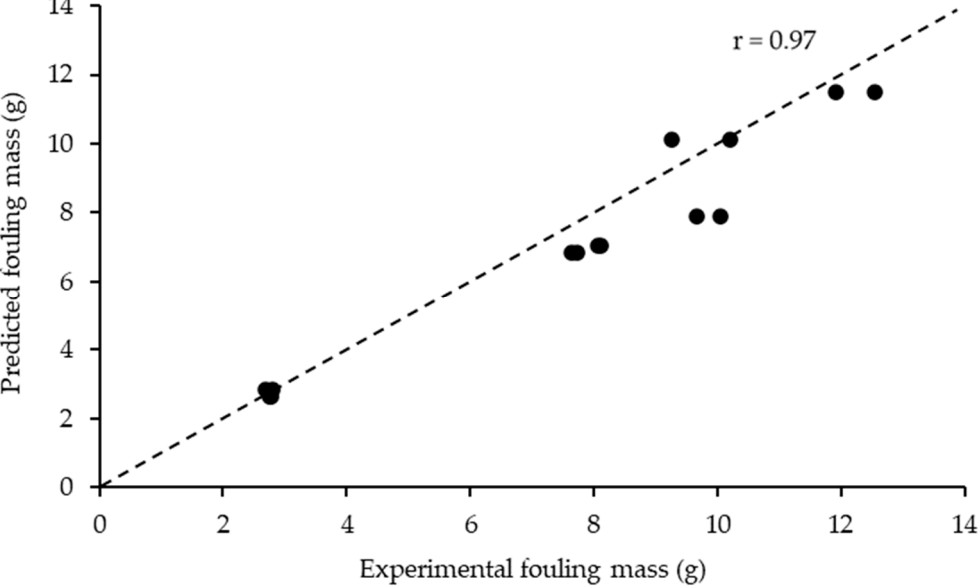

**Figure 5.** Comparison of predicted and experimental fouling deposit mass of soymilk formed on the heat exchanger plates. Experimental values were obtained with $50\,°C \leq T_{pi} \leq 55\,°C$, $70\,°C \leq T_s \leq 85\,°C$, and $65\,°C \leq T_{po} \leq 70\,°C$.

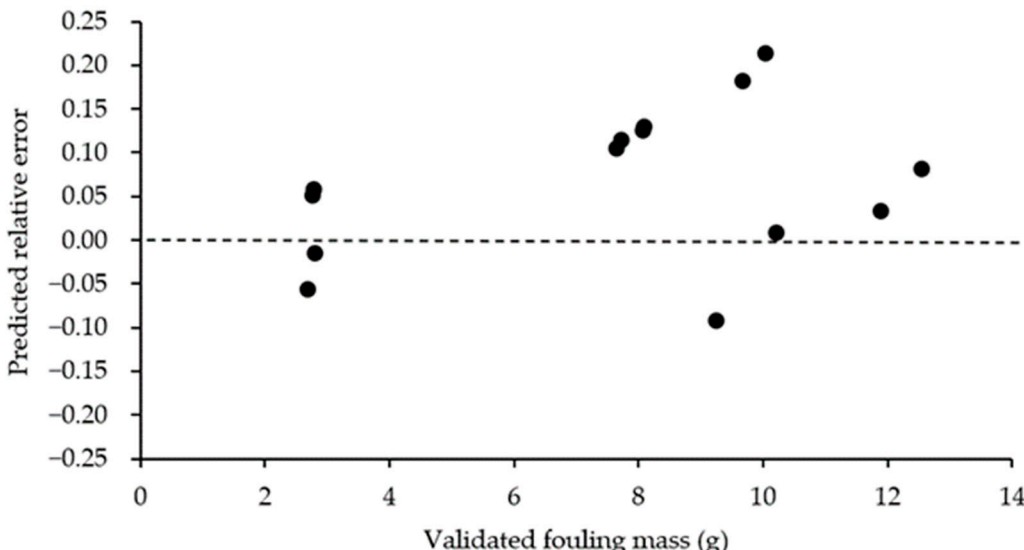

**Figure 6.** The relative error of the predicted model compared to the experimental values of soymilk fouling deposit mass. Validation data were obtained with 50 °C $\leq T_{pi} \leq$ 55 °C, 70 °C $\leq T_s \leq$ 85 °C, and 65 °C $\leq T_{po} \leq$ 70 °C.

## 4. Conclusions

Based on the dimensional analysis approach, relevant variables of fouling deposit mass could be grouped into six dimensionless terms. A simple prediction model for soymilk fouling deposit mass ($m_f$) formed on a plate heat exchanger was developed by focusing on product and surface temperatures. The data used in the development of the model were obtained from process conditions with the product inlet temperature ($T_{pi}$), the product outlet temperature ($T_{po}$), and plate surface temperature in the ranges of 50–55 °C, 65–70 °C, and 70–85 °C, respectively. A high correlation coefficient of 0.97 with an average relative error of 9.03% was observed from the plot between predicted and experimental value of fouled mass. The findings suggest the capability of a simple dimensionless model to estimate soymilk fouling deposit mass within the variables ranges studied.

**Author Contributions:** Conceptualization, E.S. and N.N.; methodology, E.S., N.N., T.S. and G.S.; software, E.O.; validation, E.S., N.N. and E.O.; formal analysis, E.S. and N.N.; investigation, N.N.; resources, E.O. and T.S.; data curation, E.O. and J.C.; writing—original draft preparation, E.S., N.N. and G.S.; writing—review and editing, E.S., N.N. and G.S.; visualization, T.S. and J.C.; supervision, E.S. and N.N.; project administration, E.O., J.C. and T.S.; funding acquisition, N.N. All authors have read and agreed to the published version of the manuscript.

**Funding:** This work was financially supported by the School of Engineering, King Mongkut's Institute of Technology Ladkrabang, Research Fund under contract number 2562-02-01-032.

**Institutional Review Board Statement:** Not applicable.

**Informed Consent Statement:** Not applicable.

**Data Availability Statement:** Not applicable.

**Conflicts of Interest:** The authors declare no conflict of interest.

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
