# Peer review of "Development of Mathematical Model to Predict Soymilk Fouling Deposit Mass on Heat Transfer Surfaces Using Dimensional Analysis"

_computation, doi:10.3390/computation11040083_

Round 1

Reviewer 1 Report

This manuscript deals with the investigation of the variables that influence the soymilk fouling deposit on contact surfaces of a plate heat exchangers. Different conditions were investigated, and a simple prediction model of soymilk fouled mass was developed using the dimensional analysis approach.

The reviewer thinks that the work is interesting, but needs some improvements before been published.

Comments

·        Could the model developed be used for other type of milk or in general fluids? (For sure after a re-evaluation of the parameters). Please comment this aspect.

·        It is not clear to the reviewer if the innovation of this paper is the methodology of the model applied or just the application of such methodology on the soymilk heating.

·        There are reference lumps in the text such as: [2,3,4]. Please eliminate this lump. After that please check the manuscript thoroughly and eliminate ALL the lumps in the manuscript. This should be done by characterizing each reference individually. This can be done by mentioning 1 or 2 phrases per reference to show how it is different from the others and why it deserves mentioning.

·        Line 47-48: Which kind of “sophisticate engineering analysis” is needed?

·        The reviewer suggests, for the introduction, to provide a summary table in which the authors can report all the model of the scientific literature with their advantages and disadvantages. In this way, will be clearer for the readers that could also immediately compare the different model between each other’s.

·        Line 158: Are you sure that you can fix density and viscosity value? Don't these values change with temperature? Please comment about this aspect because you make some tests also at higher temperature than 65 °C.

·        Line 182: Please explain better the “Buckingham’s pi- theorem”, because it is also not much clear how you can evaluate the different parameters starting from table 2 (line 191, equation 2)

·        Line 218-220: why these conditions and not others? (Please comment this aspect)

·        Line 228-230: some comment as the previous one.

·        Figure 2: The reviewer doesn't understand why the figure is made just like that. why these angles for some dimensionless numbers? Please make a comment, and only if it possible, please make the figure clearer.

·        Equation 7: There isn’t the reference of this equation in the test

·        Line 297: Equation 9 doesn’t exist.

·        Figure 4: The reviewer thinks that including in the figure the operating temperatures near to each plate (and not in the caption) could lead to a clearer figure for the reader. 

Reviewer 2 Report

The manuscript from Sritham et al. describes model to predict the soymilk fouling deposit mass using dimensional analysis. The article is correctly written and falls within the scope of the journal.

However, it has a few flaws that should be removed:

1. The model is very simple, because most of the six dimensionless numbers have been calculated as constants that are specific to the designed experiment. After the simplifications applied, the model describes only the influence of three variables on the deposition of fouling mass on the heat exchanger plates, which significantly limits its usefulness.

2. The model does not fit the results. In the case of p3= 2 it fails completely. It is easy to see in Figure 3.

3. The same remark applies to the correlation coefficient R2.

4. It should be explained why the point p3= 2 differs from the other measurements

5. The number of measurement points is too small to derive the regression equation.

Reviewer 3 Report

Please see that attached report

Round 2

Reviewer 1 Report

The authors have improved the paper according to the reviewer suggestions. So the paper can be accepeted.

Author Response

Thank you so much.

Reviewer 2 Report

The authors responded to my comments, but their explanations are unsatisfactory.

1. If the process of protein aggregation begins in given temperature conditions, which significantly affects the mass of the sediment deposited on the exchanger plates, the area around this point should be thoroughly examined. It should be clarified whether there is really an extreme at or near this point, because it would be an important discovery in this work. Examination around this point should be repeated to confirm or rule out the existence of the extreme. We can't throw away a measurement point just because it doesn't suit us for further calculations. In addition, performing additional measurements would increase the number of measurement points on the basis of which the regression equation was developed.

2. The small number of measurement points is still a problem in this experiment. The linear regression equation was developed on the basis of only 4 measurement points. The high agreement of the predicted and experimental values is no explanation here. It is normal that with a small number of measurement points, the fit of the model is very good. With a smaller number of points it will be even better, with two points it will be perfect. I believe that making at least two additional measurements in the nearby of the pi-=2 point would verify the allegations.

Round 3

Reviewer 2 Report

The article can go as it.